# New Insights on the Mediating Role of Emotional Intelligence and Social Support on University Students’ Mental Health during COVID-19 Pandemic: Gender Matters

**DOI:** 10.3390/ijerph182412935

**Published:** 2021-12-08

**Authors:** Carla Barros, Ana Sacau-Fontenla

**Affiliations:** Faculty of Human and Social Sciences, University Fernando Pessoa, 4249-004 Porto, Portugal; pssacau@ufp.edu.pt

**Keywords:** mental health, emotional intelligence, social support, gender differences, university students, COVID-19

## Abstract

Due to the demanding changes caused in the population by the COVID-19 pandemic, including a persisting experience of fear and social isolation, multiple studies have focused on the protective role of several psychological characteristics on mental health. Emotional intelligence and social support are commonly linked to mental health and well-being. The present study aims to analyze the mediator role of emotional intelligence and social support on university students’ mental health, taking into consideration the role of gender differences. An online questionnaire was administered to a sample of 923 university students during the COVID-19 lockdown in Portugal. Significant gender differences were found on mental health symptoms, emotional intelligence, and social support. A double mediation model was computed to verify if gender influences on mental health were mediated by emotional intelligence and social support. The results show indirect effects of gender on mental health. However, as both mediators mediate in the opposite direction, the total indirect effects become null. Thus, a strong direct effect of gender on mental health remains. The results of the present study have theoretical implications on protective factors of mental health by gender and practical implications for psychological intervention in university counselling services.

## 1. Introduction

The pandemic associated with COVID-19 has triggered new challenges for humanity caused by a virus with a high rate of contagion. However, long-term effects are still unknown in the scientific community. The past two years have been marked by restrictions with several confinements established by all countries which have had a high economic and social impact. The social distancing measures endorsed by the World Health Organization abruptly changed social interactions and interpersonal relationships leading to different behavioral responses [1,2]. The restrictions imposed by this situation caused an increase in psychological disorders, compromising the mental health of the population [3,4,5].

This health calamity has affected emotional and social stability, risking the well-being and mental health of the population not only today but also for years to come [6,7]. Negative emotions gain expression, leading to an increase in depression, anxiety, and stress in the population [8,9,10]. Manifestations of feelings of sadness, uncertainty, fear, hopelessness, as well as disrupted sleep patterns and concentration were found in younger people [11,12]. In fact, studies focused on this group demonstrated the presence of adaptation difficulties and negative emotional states with a significant impact on their mental health [13,14].

University students were faced with a more demanding training model in the management, organization, and achievement of academic goals [15]. The extended time and the increase of distance learning and remote training have aggravated their psychological vulnerability [16,17], with a significant rise in psychological disorders [11,12,18]. Restriction measures such as quarantine, social distancing, social isolation, and obligatory mask-wearing were mandatory and brought huge changes in academic life. Nonetheless, variations in the teaching-learning model, the practical teaching and internships interrupted, adjustment of studying environment, and social isolation, among other factors, have revolutionized the daily life of university students. In fact, the risk of mental health disorders increased (in particular higher levels of anxiety, depression, and stress among university students) [3,12,19,20,21].

In this period of uncertainty, psychological stability plays a decisive role in the promotion and development of mental health, and emotional intelligence seems to play an important role in regulating and managing emotions. The relationships between emotional intelligence and mental health involve several facets of emotional intelligence, which can be protective of psychological well-being [22,23]. Therefore, people with high levels of emotional intelligence have a greater capacity to deal with the negative effects of stress [24,25,26].

The increase of research has accentuated the importance of emotional regulation for optimal health [27,28,29,30]. Several studies point to a relationship between emotional intelligence and psychological functioning. Self-esteem, social competence, and satisfaction with life seem to be higher when emotional skills are present [31,32]. Emotional skills used to identify feelings and emotions facilitate more effective emotional strategies to deal with negative events [16,17,23,24]. Moreover, emotional regulation plays an important role in adapting people to the demands of the social environment and can help overwhelm stressful situations [33,34]. Thus, it seems that emotional intelligence can increase the frequency and preservation of positive emotions and plays an effective moderator of mental health by improving social adjustment and psychological well-being [34,35].

Recent studies showed emotional intelligence as a good protector from the negative effects of stress, anxiety, and depression in COVID-19 lockdown for university students [3,36,37] and demonstrated that it was positively related to the way students manage stress [38]. However, recent studies have also drawn attention to a possible gender effect (although the results are not conclusive) [39,40,41].

Besides emotional regulation strategies, social support was identified as a protective factor across all three symptom domains of mental health [42,43,44]. More social support indicates more sense of belonging and promotes positive attitudes [45]. The support network has the function of being a minimizer factor of stress. In this sense, social support is directly related to emotional and everyday support, which involves relationships of warmth, care, and attention provided by family, friends, and significant others. This process makes the individual feel loved, cared for and safe, contributing to the feeling of coherence and control over his life [42,45].

In the uncertainty and hopelessness during the COVID-19 pandemic, support care from family and friends may have protected young adults from loneliness, fear, worries, and other negative thoughts. Therefore, perceived social support has provided university students with more social and interpersonal well-being, which leads to better psychological functioning [37,45,46,47]. Thus, social support, including family and friends, can act as a protective factor for mental disorders, as it supports the individuals and gives them the possibility of dealing with unexpected events.

This study aims to analyze the role of emotional intelligence and social support on university students’ mental health in the pandemic. We hypothesized that perceived social support is a mediator of the effects of emotional intelligence on mental health. The mediation model is represented in Figure 1. However, we also intend to understand the role that gender plays in this mediation model.

## 2. Materials and Methods

### 2.1. Participants

A sample of 923 Portuguese university students was recruited to fill out an online questionnaire. 71.2% (*n* = 657) were female and 28.8% (*n* = 266) were male. The mean age of the participants is 20.66 (*sd* = 4.265). 95.9% (*n* = 885) are single. In relation to the attendance regime applied since the beginning of the pandemic, 57.8%, (*n* = 534) followed academic activities mostly or completely at distance, 32.4% (*n* = 299) in a mixed regime (with approximately 50% of the time in person and 50% on a distance basis) and only 9.8% (*n* = 90) followed a mostly or completely face-to-face regime (see Table 1).

### 2.2. Procedure

Data were collected online by sharing a questionnaire through Google Forms between 26 March and 6 April 2021. The sampling was non-probabilistic by the snowball technique with university students from Portugal. The questionnaire entails different scales starting with a cover page with a short explanation about the aim of the study. The criteria for participation in the study were informed consent, voluntary participation, and confidentiality. Informed consent was obtained by all participants previous to answering the survey. The estimated time to complete the questionnaire was around 10 min. The present study was approved by the ethics committee of the University of Fernando Pessoa (Porto, Portugal, Ref. PI-138/21) and respect all procedures of the Declaration of Helsinki.

### 2.3. Measures

#### 2.3.1. Emotional Intelligence

The WLEIS emotional intelligence scale [48] was used to evaluate emotional intelligence using the Portuguese version [49]: It is a 16-item scale on a 5-point Likert scale (1—Strongly disagree; 5—Strongly agree). This scale measures four dimensions of emotional intelligence: (i) the evaluation and expression of emotions themselves, (ii) evaluation and recognition of emotions in others, (iii) regulation of emotions of one’s own, and (iv) use of emotions.

#### 2.3.2. Mental Health

The DASS-21, depression, anxiety and stress scale [50] was used to assess the symptoms of depression, anxiety, and stress using the Portuguese version [51]. This scale is composed of 21 items, 7 for each 3factors with a 4-point Likert scale (0: did not apply to me at all; 3—applied to me very much or most of the time). Higher scores in DASS-21 correspond to higher levels of anxiety, depression, and stress. The degree of severity of symptoms in each dimension is obtained via the sum of the scores of the answers to the items corresponding to each of the factors. To obtain the final score on the original scale from 0 to 42 points it is necessary to multiply each factor by two. So, for depression, 0 to 9 is considered normal, 10 to 13 is mild, 14 to 20 is moderate, 21 to 27 is severe and more than 28 is very severe. For anxiety, 0 to 7 is normal, 8 to 9 is mild, 10 to 14 is moderate, 15 to 19 is severe and more than 20 is very severe. For stress, 0 to 14 is normal, 15 to 18 is mild, 19 to 25 is moderate, 26 to 33 is severe and more than 33 is very severe (Lovibond and Lovibond, 1995).

#### 2.3.3. Social Support

The MSPSS, multidimensional scale of perceived social support [52,53] evaluates an individual’s perception of social support using the Portuguese version [54]. This scale consists of three categories of social support: family, friends, and significant others in the 12-item version. Each of the 3 subscales (family, significant others, and friends) consists of 4 items scored on a 7-point Likert scale from 1 (very strongly disagree) to 7 (very strongly agree). The overall score is the outcome of the 12 items.

#### 2.3.4. Demographics

Participants’ demographic data, collected with closed questions, included information about age, gender, and marital status. It also comprised a question regarding class attendance regime during the pandemic.

### 2.4. Statistical Analyses

We performed descriptive statistics for sample description on all variables assessed. We also performed Anova for gender differences with partial eta squared for the estimation of size effects of gender on mental health, emotional intelligence and social support and Pearson’s correlation for relations among variables. In order to test the mediation model, we used the PROCESS Macro for SPSS, model 6. Data analysis was conducted using IBM SPSS Statistics version 26 (SPSS Inc.: Chicago, IL, USA). All analyses were computed with 95% confidence interval and *p* < 0.05.

## 3. Results

### 3.1. Preliminary Analysis

Correlation analysis shows a significant relation between emotional intelligence, social support and mental health. Pearson’s r is shown in Table 2. All four dimensions of emotional intelligence show a positive correlation with all dimensions of social support (family, friends, and others). Both emotional intelligence (except others’ emotions appraisal) and social support (except no significant relation between stress and significant others’ support) show a negative relation with mental health.

Anova comparisons confirm significant differences between male and female in all variables in the analysis (Table 3). Male students show higher levels of mental health with lower scores in anxiety, depression and stress symptoms. Male students also showed significant higher levels of emotional intelligence while female students exhibited higher levels of social support.

These results led us to include gender as a key factor on the mediation model presented above (see Figure 1).

There was no significant effect of attendance (distance, presence and 50%) on mental health (F(2) = 0.98, *p* > 0.05) or on social support (F(2) = 1.83, *p* > 0.05).

### 3.2. Mediation Effects

Our initial hypothesis suggested that in a pandemic context with confinement of populations the effect of emotional intelligence on university students’ mental health is mediated by social support. However, preliminary data analysis shows that gender must be seen as a key factor. Significant differences between women and men in IE, SS, and all mental health symptoms (anxiety, depression, and stress) lead us to restructure the mediation model in which gender (VI) has a priority role in explaining the effects of EI (M1) and SS (M2) on university students’ mental health (VD). Then a double mediation model is proposed. Computing analysis results in four regression equations and three indirect effects. The results of the double mediation model show significant direct and indirect effects among all variables in the analysis. Significant direct effects are shown in Figure 2.

Gender (0 = male, 1 = female) show a negative direct effect on emotional intelligence (b = −1.518, se = 0.698, t(921) = −2.175, *p* = 0.029), a positive direct effect on social support (b = 4.013, se = 0.991, t(920) = 4.048, *p* = 0.000), and a positive direct effect on anxiety, depression and stress (b = 6.432, se = 0.914, t(921) = 7.036, *p* = 0.000). Female students show less emotional intelligence, higher levels of social support, and a higher prevalence of anxiety, depression and stress symptoms. Moreover, both mediators (emotional intelligence and social support) have a direct negative effect on anxiety, depression, and stress (b = −0.538, se = 0.045, t(919) = −11.853, *p* = 0.000 and b = −0.236, se = 0.030, t(919) = −7.829, *p* = 0.000, respectively). Students with higher levels of emotional intelligence and social support show less symptoms of anxiety, depression, and stress. Finally, emotional intelligence shows a positive direct effect on social support (b = 0.513, se = 0.047, t(920) = 10.991, *p* = 0.000).

Indirect effects were also found. As shown in Table 4, all mediation paths are significant so gender is influencing mental health via emotional intelligence and social support (Path 1: b = 0.817, BootSE = 0.424, 95% CI [0.042, 1.702]; Path 2: b = −0.947, BootSE = 0.290, 95% CI [−1.55, −0.42]; Path 3: b = 0.184, BootSE = 0.096, 95% CI [0.009, 0.390]). However, results confirm a partial mediation model, and not a complete one, since c’ coefficient (b = 6.432) is greater than 0 showing that gender has strong direct effects on mental health. Despite all the indirect paths being significant, total indirect effect is not (b = 0.054, BootSE = 0.591, 95% CI [−1.077, 1.244] since the indirect effects are of the opposite sign. As shown in Table 4, path 1 and path 2 show the strongest effects with opposite b coefficients (0.817 and −0.947, respectively). As the total indirect effect is the sum of all indirect effects (path 1 + path 2 + path 3) indirect effects cancel each other out.

## 4. Discussion

The COVID-19 lockdown increased psychological vulnerability in Portuguese university students and significantly affected students’ mental health. Our data reveal that more than a half of students present anxiety, depression, and stress symptoms. Of particular concern are the results on anxiety with 25% of students showing extremely severe symptoms. Transition to online learning, peer social isolation, a lack of leisure activities, and the impossibility of experiencing student contexts interactions may have been especially demanding for the university students. However, our results show that attending lectures in-person, remotely, or in a mixed regime had no effects on mental health or on social support. We understand that these results show that even those who attended classes in person or in a mixed regime (50% in-person, 50% remotely) had very limited social contacts at the university. At that time, canteens, gyms, and other social contexts for students were closed; students had no possibility of developing group works and they had the obligation to maintain social distance so they did not benefit from any advantage for being present in class. The prevalence of mental disorders found at the present study requires psychological attention, in agreement with previous studies [12,44,55]. Gender differences in mental health are relevant, showing a greater vulnerability in women compared to men. As other studies also show, during COVID-19 lockdown females report more negative impact regarding stress, anxiety and depression [56,57,58,59,60]. These differences are also confirmed in previouspre-pandemic studies, which might indicate an overall vulnerability among females when compared to male students. [41,61,62,63,64].

The mediation model shows new insights on the role of gender on mental health through emotional intelligence and social support. Our results suggest that gender differences in mental health are reinforced by gender differences in emotional intelligence. Emotional intelligence plays an important role in adapting people to the demands of the social environment, helping university students to overcome stressful situations [34,36,38]. Although several previous studies about gender differences showed that females seem to recognize and deal with emotions better than males [22,39,40,65,66,67], recent studies during the COVID-19 pandemic reveal that males scored significantly higher than females on several emotional intelligence such as emotion management and emotional stability [23,41,68].

In accordance with other studies [69,70,71,72,73], our study shows that male students have higher levels of emotional intelligence than female students. It also shows that emotional intelligence is a good predictor of mental health, having a direct negative effect on the presence of symptoms of anxiety, depression, and stress [37,61,74].

On the other hand, the model also suggests that social support plays an important mediating role in the effect of gender on mental health. In this case, female students report greater social support during the pandemic than male students. As social support also works as a direct negative predictor of mental health symptoms, females’ higher levels of social support are translated into a decrease in symptoms of anxiety, depression, and stress. The important role of social support during COVID-19 pandemic has already been mentioned by previous studies with university students [14,38,47,58]. Consistently, these studies reveal that being connected with and supported by friends and family members during the COVID-19 pandemic worked as a coping mechanism, this being a protective factor of mental health [44]. In the uncertainty, anxiety, and despair during the COVID-19 pandemic, dispositional care from family and friends may protect young adults from worries, rumination thoughts, fear, and loneliness. Our study suggests that female students benefited more from this protective role.

However, since the indirect effects of gender on mental health show the opposite sign, the role of emotional intelligence and social support for both genders is nullified. Thus, and in the case of female students who conform to the group with the highest levels of anxiety, depression, and stress, the protective role of social support is nullified by the adverse effect of showing lower scores in emotional intelligence. In the case of male students, lower levels of social support minimize the protective effect of emotional intelligence, a variable in which they score higher than their female colleagues. For this reason, and although emotional intelligence and social support are important protector factors in times of a pandemic [44,47,57], the effect of gender must be considered as also having a strong direct effect. These strong gender effects on mental health led us to question how gender-related risk factors may have an effect during the disruption of everyday life. New studies must be carried out on how female and male students cope with stressors and the role of risk and protective factors.

The present study has some limitations. The impact of the COVID-19 pandemic on mental health was assessed using a self-report measure whereby participant’s responses may have been affected by their mood states at the time. Longitudinal studies must be carried out to evaluate the long-term impact of the COVID-19 pandemic.

## 5. Conclusions

Our findings provide relevant insights that may have useful theoretical and practical implications. From the theoretical perspective, further research is needed to deepen the analysis of gender differences. It is important to confirm whether the results of the present study are showing that the pandemic situation (or other adverse events) can have a very different impact on both sexes, changing competences, skills, and behavior patterns or if we are looking at a generational evolution on gender differences. In response to the pandemic, many universities abruptly suspended regular classroom sessions as well as other academic and social activities associated to university life which led to a variety of negative social and psychological consequences for university students. Those psychological effects are being widely reported by scientific literature. Emotional intelligence, as well as social support, plays an important role for coping with adversity but a gender perspective must be applied to understand and intervene.

These findings also hold implications for university counselling services during especially stressful periods for the students. Results show the importance of developing continuous psychological interventions among students and evaluating their mental health and well-being over time. Our data show a trend that aligns with other university student populations assessed in several countries. In sum, these results emphasize the importance of developing global and gendered interventions that include the overall differences of university students.

## Figures and Tables

**Figure 1 ijerph-18-12935-f001:**
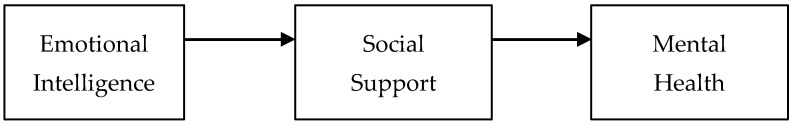
The mediation role of social support (SS) on the emotional intelligence (EI) effects on mental health (MH).

**Figure 2 ijerph-18-12935-f002:**
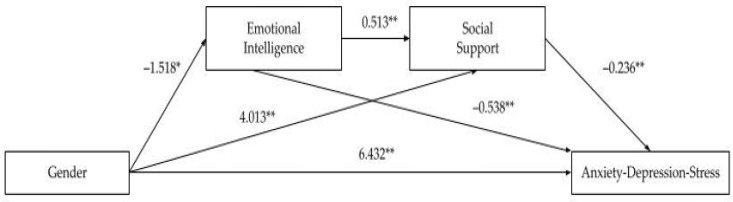
Significant direct effects of the double mediation model (b-Coefficients). * *p* < 0.001; ** *p* < 0.0001.

**Table 1 ijerph-18-12935-t001:** Sample demographic and mental health symptoms description.

Variable	%					Mean (sd)
Gender						
Female	71.2					
Male	28.8					
Age						20.66 (4.265)
Marital Status						
Single	95.9					
Married	3.6					
Other	0.6					
Attendance						
Distance	32.4					
Both (almost 50%)	57.8					
Face-to-face	9.8					
MH Symptoms	Normal	Mild	Moderate	Severe	Extremely severe	
Anxiety	42.5	6.7	17.6	8.2	25.0	5.920 (5.173)
Depression	39.5	12.8	20.7	10.1	16.9	7.281 (5.553)
Stress	44.0	12.8	18.2	16.5	8.6	8.792 (5.271)

**Table 2 ijerph-18-12935-t002:** Pearson’s correlation coefficients among EI, SS, and MH dimensions.

	2	3	4	5	6	7	8	9	10
1. MH-Anx	0.69 ***	0.82 ***	−0.37 ***	0.04	−0.24 ***	0.38 ***	−0.33 ***	−0.21 ***	−0.11 **
2. MH-Dep		0.74 ***	−0.41 ***	−0.03	−0.44 ***	−0.37 ***	−0.46 ***	−0.31 ***	−0.19 ***
3. MH-Str			−0.39 ***	0.06	−0.28 ***	−0.48 ***	−0.30 ***	−0.21 ***	−0.05
4. EI-Self				0.21 ***	0.43 ***	0.52 ***	0.26 ***	0.15 ***	0.18 ***
5. EI-Others					0.25 ***	0.18 ***	0.10 **	0.18 ***	0.22 ***
6. EI-Use						0.41 ***	0.30 ***	0.23 ***	0.15 ***
7. EI-Control							0.20 ***	0.21 ***	0.10 **
8. S-Family								0.48 ***	0.44 ***
9. S-Friends									0.53 ***
10. S-Others									

Note: MH-Anx = DASS-21 Anxiety Subscale; MH-Dep = DASS-21 Depression Subscale; MH-Str = DASS-21 Stress Subscale; EI-Self = Self-Emotional Appraisal; EI-Others = Others’ Emotion Appraisal; EI-Use = Use of Emotions; EI-Control = Regulation of Emotions; S-Family = MSPSS Family Subscale; S-Friends = MSPSS Friends Subscale; S-Others = MSPSS Significant Others Subscale. ** *p* < 0.01; *** *p* < 0.001.

**Table 3 ijerph-18-12935-t003:** Gender differences of mental health, emotional intelligence and social support.

Dimensions	Gender	Mean (sd)	F	*p*	Partial Eta^2^
MH-Anx	Female	6.50 (5.31)	29.485	0.000	0.031
	Male	4.49 (4.52)			
MH-Dep	Female	7.70 (5.63)	12.894	0.000	0.014
	Male	6.26 (5.23)			
MH-Str	Female	9.67 (5.15)	67.286	0.000	0.068
	Male	6.63 (4.94)			
MH-Total	Female	23.86 (14.70)	38.909	0.000	0.041
	Male	17.38 (13.27)			
EI-Total	Female	55.17 (9.28)	4.732	0.030	0.005
	Male	56.69 (10.34)			
SS-Total	Female	68.71 (13.76)	9.464	0.002	0.010
	Male	65.47 (16.07)			

MH-Anx = DASS-21 Anxiety Subscale; MH-Dep = DASS-21 Depression Subscale; MH-Str = DASS-21 Stress Subscale; MH-Total = DASS-21 Total Score; EI-Total = WLEIS Total Score; SS-Total = MSPSS Total Score.

**Table 4 ijerph-18-12935-t004:** Significant indirect, direct and total effects (b-Coefficients and 95% CIs) of gender (IV), emotional intelligence (M1), and social Support (M2) on mental health (DV).

Mediation Effects	b-Coefficients and 95% CIs
Indirect effects:	
Path 1: Gender → EI → MH Symptoms	0.817 [0.042, 1.702]
Path 2: Gender → SS → MH Symptoms	−0.947 [−1.555, −0.421]
Path 3: Gender → EI → SS → MH Symptoms	0.184 [0.009, 0.388]
Total indirect effects	0.054 [−1.077, 1.244]
Direct effects (c’ coefficient): Gender → MH Symptoms	6.432 [4.638, 8.226]
Total Effects (c coefficient)	6.486 *

EI = Emotional Intelligence; MH = Mental Health; SS = Social Support. * *p* < 0.000.

## Data Availability

The data presented in this article are not readily available since they were not approved to be shared outside of the research team. Requests to access the datasets should be directed to cbarros@ufp.edu.pt or pssacau@ufp.edu.pt.

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
