# Peer review of "New Insights on the Mediating Role of Emotional Intelligence and Social Support on University Students’ Mental Health during COVID-19 Pandemic: Gender Matters"

_ijerph, 2021, doi:10.3390/ijerph182412935_

Round 1

Reviewer 1 Report

To make this article more significant and of interest to readers, it may be interesting to mention how your study results are different than similar pre-pandemic studies - such a comparison may add value to your manuscript.

Please review English language and style and make appropriate changes (for example, in section 2.3.3 Social support, you write : "(...) by the summed up of each item" which must be rewritten. There are many such examples in your paper.

Please write out "10 minutes" in section 2.2 procedure

In section 2.3.2 Mental Health, you mention both DASS-21 and EADS-21 Are these two different scales?

 In Table 2, correlation between 1. MH-Anx and EI-Control is .,38*** please make correction 

Please fix arrows in Figure 2

In section 3.2 mediation effects, you mention the effects of "IE" instead of "EI". Please correct.

In Table 4. the total indirect effect that is revealed in your results section is not indicated in the table

In section 3.2. mediation effects, you also mention "because the indirect effects are of the opposite sign and, therefore, the mediating effects are annulling each other" - are you referencing the suppression effect? If so, please reformulate and add a source. I also suggest that you explain this mediation result in your discussion. What may explain this result? How does this result impact the validity of your study? Have similar results been found in previous (pre-pandemic) studies?

I also recommend that you further explain the theoretical and practical implications of your study results. 

Author Response

Thank you for the Comments and Suggestions 

To make this article more significant and of interest to readers, it may be interesting to mention how your study results are different than similar pre-pandemic studies - such a comparison may add value to your manuscript.

We have improved the discussion including a comparation with similar pre-pandemic studies.

Please review English language and style and make appropriate changes (for example, in section 2.3.3 Social support, you write: "(...) by the summed up of each item" which must be rewritten. There are many such examples in your paper.

The manuscript was revised by an English native.

Please write out "10 minutes" in section 2.2 procedure

We have included in the improved version.

In section 2.3.2 Mental Health, you mention both DASS-21 and EADS-21 Are these two different scales?

We have included DASS-21 in the improved version.

 In Table 2, correlation between 1. MH-Anx and EI-Control is .,38*** please make correction

We have made the correction in the improved version.

Please fix arrows in Figure 2

We have included in Figure 2 the JPEG Format to fix the arrows in the improved version.

In section 3.2 mediation effects, you mention the effects of "IE" instead of "EI". Please correct.

We have made the correction in the improved version.

In Table 4. the total indirect effect that is revealed in your results section is not indicated in the table

We have included the total indirect effect in Table 4 in the improved version.

In section 3.2. mediation effects, you also mention "because the indirect effects are of the opposite sign and, therefore, the mediating effects are annulling each other" - are you referencing the suppression effect? If so, please reformulate and add a source. I also suggest that you explain this mediation result in your discussion. What may explain this result? How does this result impact the validity of your study?

We appreciate this comment which made us to read about suppression effect. After some reading (Agler, R. & De Boeck, P. 2017; MacKinnon, Krull & Lockwood, 2000; Murgui, S. & Jiménez, T.I., 2013) we came to the conclusion that the effect reflected in our results is, in fact, a partial mediation effect and not a suppression effect. All authors consulted describe the suppression effect in relation to the relationship between indirect and direct effects. Following their arguments, there will be a suppression effect if, summarizing, c < c’ that is not our case. We have not found any paper or work in which the suppression effect was interpreted in relation to how indirect effects affect each other in a double mediation model (what we express in our paper as: “indirect effects are of the opposite sign and, therefore, the mediating effects are annulling each other”. We changed the sentence to better express our interpretation.

We really appreciate any extra comment about our rationale.

Main readings:

Agler, R. & De Boeck, P. (2017). On the Interpretation and Use of Mediation: Multiple Perspectives on Mediation Analysis. Frontiers in Psychology, 8, 1984. https://doi.org/10.3389/fpsyg.2017.01984

MacKinnon, D.P., Krull, J.L., & Lockwood, C.M. (2000). Equivalence of the Mediation, Confounding and Suppression Effect. Prevention Research, 1(4), 173-181. https://dx.doi.org/10.1023%2Fa%3A1026595011371 Author manuscript available in PMC 2010 Feb 10 at: https://www.ncbi.nlm.nih.gov/pmc/articles/PMC2819361/pdf/nihms-173346.pdf

Murgui, S. & Jiménez, T.T (2013). Efecto de supresión y mediación en el contexto de la intervención psicosocial: Diferencias, similitudes y ejemplos. Psychosocial Intervention, 22, 55-59. http://dx.doi.org/10.5093/in2013a7

Have similar results been found in previous (pre-pandemic) studies?

We have improved the discussion including a comparation with similar pre-pandemic studies.

I also recommend that you further explain the theoretical and practical implications of your study results.

We have improved the discussion/conclusions.

Reviewer 2 Report

The study aims to analyze the role of emotional intelligence and social support on university students’ mental health in a pandemic time. And the authors hypothesized that perceived social support is a mediator of the effects of emotional intelligence on mental health and that the gender also plays a role as a mediator.

The theoretical background of the study is very well written, and the authors have done a good literature review.

However, the authors state that ‘’In relation to the attendance regime applied since the beginning of the pandemic, 57.8%, (n = 534) followed academic activities mostly or completely at distance, 32.4% (n = 299) in a mixed regime (with approximately 50% of the time in person and 50% on a distance basis) and only 9.8% (n = 90) followed a mostly or completely face-to-face regime’’.

And in the discussions, there is a sentence: ‘’Transition to online learning, peer social isolation, a lack of leisure activities, and the impossibility of experiencing student contexts interactions may have been especially demanding for the university students’’, but the authors didn’t assess in any way the impact on online learning on any of the variables assessed in the study. Base on their data we can assume that we have 3 categories of students who follow 3 different type of attendance regime, and we can assume they have different levels of interaction with theirs peers (the one in mixed regime or in person should have a different level of social support?).

So I would like to see some statistical analysis that look also at the attendance regime, with the difference (if any) in scores for mental health (anxiety, depression, stress and social support) analyzed.

And in functions of the new results presented I suggest the authors rewrite the discussions and conclusions.

Author Response

Thank you for the comments and suggestions 

The study aims to analyze the role of emotional intelligence and social support on university students’ mental health in a pandemic time. And the authors hypothesized that perceived social support is a mediator of the effects of emotional intelligence on mental health and that the gender also plays a role as a mediator.

The theoretical background of the study is very well written, and the authors have done a good literature review.

However, the authors state that ‘’In relation to the attendance regime applied since the beginning of the pandemic, 57.8%, (n = 534) followed academic activities mostly or completely at distance, 32.4% (n = 299) in a mixed regime (with approximately 50% of the time in person and 50% on a distance basis) and only 9.8% (n = 90) followed a mostly or completely face-to-face regime’’.

And in the discussions, there is a sentence: ‘’Transition to online learning, peer social isolation, a lack of leisure activities, and the impossibility of experiencing student contexts interactions may have been especially demanding for the university students’’, but the authors didn’t assess in any way the impact on online learning on any of the variables assessed in the study. Base on their data we can assume that we have 3 categories of students who follow 3 different type of attendance regime, and we can assume they have different levels of interaction with theirs peers (the one in mixed regime or in person should have a different level of social support?).

So I would like to see some statistical analysis that look also at the attendance regime, with the difference (if any) in scores for mental health (anxiety, depression, stress and social support) analyzed.

And in functions of the new results presented I suggest the authors rewrite the discussions and conclusions.

We included statistical analysis at the attendance regime. We also improved the results/discussion.

Round 2

Reviewer 2 Report

The authors have revised the paper according to the suggestions and I consider the modifications have improved the paper.